# A Causal View on Robustness of Neural Networks

**Cheng Zhang** *
Microsoft Research
Cheng.Zhang@microsoft.com

**Kun Zhang**
Carnegie Mellon University
kunz1@cmu.edu

**Yingzhen Li** *
Microsoft Research
Yingzhen.Li@microsoft.com

## Abstract

We present a causal view on the robustness of neural networks against input manipulations, which applies not only to traditional classification tasks but also to general measurement data. Based on this view, we design a deep causal manipulation augmented model (deep CAMA) which explicitly models possible manipulations on certain causes leading to changes in the observed effect. We further develop data augmentation and test-time fine-tuning methods to improve deep CAMA's robustness. When compared with discriminative deep neural networks, our proposed model shows superior robustness against unseen manipulations. As a by-product, our model achieves disentangled representation which separates the representation of manipulations from those of other latent causes.

## 1 Introduction

Deep neural networks (DNNs) have great success in many real-life applications; however, they are easily fooled even by a tiny amount of perturbation [47, 17, 6, 4]. Lack of robustness hinders the application of DNNs to critical decision making tasks such as uses in healthcare. To address this, one may suggest training DNNs with diverse datasets. Indeed, data augmentation and adversarial training have shown improvements in both the generalization and robustness of DNNs [25, 37, 30]. Unfortunately, this does not address the vulnerability of DNNs to *unseen* manipulations, e.g. as shown in Figure 1, a DNN trained on clean MNIST digits fails to properly classify shifted digits. Although observing perturbations of clean data in training improves robustness against that particular manipulation (the green line), the DNN is still fragile when unseen manipulations are present (orange line). As it is unrealistic to augment the training data towards all possible manipulations that might occur, a principled method that fundamentally improves the robustness is much needed.

On the other hand, human perception is robust to such perturbations thanks to the capability of *causal reasoning* [36, 18]. After learning the concept of an "elephant", a child can identify the elephant in a photo taken under any lighting condition, location, etc. In the causal view, the lighting condition and the location are causes of the presented scene, which can be intervened without changing the presence of the elephant. However, discriminative DNNs do not take such possible interventions into account, and cannot adapt the predictor for new data gathered with unseen manipulation.

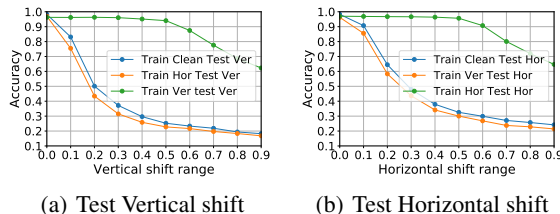

(a) Test Vertical shift    (b) Test Horizontal shift

Figure 1: Robustness of DNNs on shifted MNIST. Panels (a) and (b) show the accuracy on classifying noisy test data generated by shifting the digits vertically (Ver) and horizontally (Hor). It shows that data augmentation during training makes generalization to unseen shifts worse (orange vs blue lines).

---

In light of the above observation, we discuss the robustness of DNNs from a causal perspective. Our contributions are:

- **A causal view on robustness of neural networks.** We argue from a causal perspective that input perturbations are generated by their unseen causes that are *artificially manipulatable*. Therefore DNN's robustness issues to these input perturbations is due to the lack of causal understanding.

- **A causal inspired deep generative model.** We design a causal inspired deep generative model which takes into account possible interventions on the causes in the data generation process [50]. Accompanied with this model is a test-time inference method to learn unseen interventions and thus improve classification accuracy on manipulated data. Compared to DNNs, experiments on both MNIST and a measurement-based dataset show that our model is significantly more robustness to unseen manipulations.

## 2 A Causal View on Robustness of Neural Networks

Discriminative DNNs may not be robust to manipulations such as adversarial noise injection [17, 5, 4], rotation, and shift. They simply trust the observed data and ignore the constraints of the data generating process, which leads to overfiting to nuisance factors and makes the classification output sensitive to such factors. By exploiting the overfit to the nuisance factors, an adversary can easily manipulate the inputs to fool discriminative DNNs into predicting the wrong outcomes.

On the contrary, human can easily recognize an object in a scene and be indifferent to the variations in other aspects such as background, viewing angle, the presence of a sticker to the object, etc. More importantly, human recognition is less affected even by drastic perturbations on a number of factors underlying the observed data, e.g. variations in the lighting condition. We argue that the main difference here is due to our ability to perform *causal reasoning*, which identifies factor that are not relevant to the recognition results [13, 38, 34]. This leads to robust human perception to not only a certain type of perturbations, but also to many types of unseen manipulations on other factors. Thus we argue that one should incorporate the causal perspective into model design, and make the model robust on the level of different types of manipulations.

Before presenting our causally informed model, we first define the generative process of perceived data. There might exist multiple causes in the data generation process influencing the observed data $X$, and we visualize exemplar causal graphs in Figure 2 with the arrows indicating causal associations. Among these causes of $X$, $Y$ is the target to be predicted, $M$ is a set of variables which can be intervened artificially, and $Z$ represents the rest of the causes that cannot be intervened in the application context. Take hand-written digit classification for example, $X$ is the image and $Y$ is the class label. The appearance of $X$ is an effect of the digit number $Y$, latent causes $Z$ such as writing styles, and possible manipulations $M$ such as rotation or translation.

We can thus define valid perturbations of data through the lens of causality. Datasets are produced by interventions in general, so defining a valid attack is equivalent to defining a set of variables in the causal graph (Figure 2) which can be intervened by the adversary. We argue that *a valid perturbation is an intervention on $M$ which, together with the original $Y$ and $Z$, produces the manipulated data $X$*. In this regard, recent adversarial attacks as perturbations of the inputs can be considered as a specific type of intervention on $M$ such that a learned predictor is deceived. Note here we do not consider interventions on $Y$ and $Z$ (and their causes): interventions on $Y$ (and its causes) changes the "true" value of the target and do not correspond to the type of perturbations we are considering; by definition $Z$ (and its causes) cannot be intervened *artificially* (e.g. genetic causes are often difficult to intervene), thereby unavailable to the adversary. This also shows the importance of separating the (unobserved) causes $Z$ and $M$, as it helps to better identify the interventions presented in perturbed inputs, which also leads to improved classification robustness.

In light of the above definition on valid perturbations, it is clear that performing prediction adaptive to the (unknown) intervention is necessary to achieve robustness to manipulated data. A natural way to build such adaptive predictor is to construct a model that perform reasoning in a way consistent to the causal process. To see this, note that a valid perturbation changes the value of $M$, but it leaves the functional relationship from $M$ and $Y$ to $X$ intact. This is known as *modularity property* [50], and in this sense the causal system is autonomous [35]. Therefore a causally consistent predictive model is

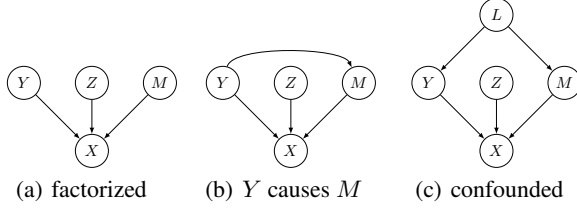

(a) factorized  (b) $Y$ causes $M$  (c) confounded

Figure 2: Exampler causal graphs with $Y$, $Z$, $M$ causing $X$. $Y$ might cause $M$ (panel b), or they might be confounded (panel c).

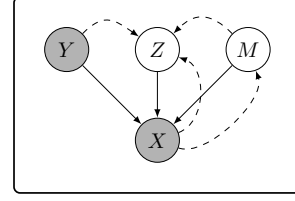

Figure 3: Graphical presentation of deep CAMA for single modal data.

expected to be able to learn this functional relationship from data, and adapt the prediction result of target in test time according to its reasoning on the underlying causal factors.

## 3 The Causal Manipulation Augmented Model

We propose a deep CAusal Manipulation Augmented model (deep CAMA), which takes into account the causal relationship for model design. We also design a fine-tuning algorithm to enable adaptive reasoning of deep CAMA for unseen manipulations on effect variables. The robustness can be further improved by training-time data augmentation, without sacrificing the generalization ability to unseen manipulations. Below we first present the deep CAMA for single modality data, and then present a generic deep CAMA for multimodality measurement data.

### 3.1 Deep CAMA for single modality data

The task of predicting $Y$ from $X$ covers a wide range of applications such as image/speech recognition and sentiment analysis. Normally a discriminative DNN takes $X$ as input and directly predicts (the distribution of) the target variable $Y$. Generative classifiers, on the other hand, build a generative model $Y \rightarrow X$, and use Bayes' rule for predicting $Y$ given $X$: $p(y|x) = p(y)p(x|y)/p(x)$.

We design deep CAMA (Figure 3) following the causal graph in Figure 2(a), which returns a factorized model:

$$p_\theta(x, y, z, m) = p(m)p(z)p(y)p_\theta(x|y, z, m). \quad (1)$$

Notice that we do not consider modelling dependencies between $Y$ and $M$ even when the causal relationship might exist and 2(c)) in the generation process of the training data (see Figures 2(b). By our definition of valid perturbation, $M$ is intervened on (i.e., $do(m)$), which blocks the influence from $Y$ to $M$, and the generation process of manipulated data reduces to the factorized case (Figure 2(a)).

For efficient posterior inference we use *amortization* [24, 39, 51] to define an inference network:

$$q_\phi(z, m|x, y) = q_{\phi_1}(z|x, y, m)q_{\phi_2}(m|x). \quad (2)$$

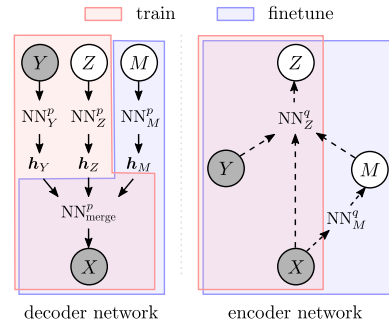

Figure 4: The network architecture. Shaded areas show the selective part for $do(m)$ training and the fine-tune method, respectively.

Here the variational parameters are $\phi = \{\phi_1, \phi_2\}$, where $\phi_1$ is network parameter for the variational distribution $q_{\phi_1}(z|x, y, m)$, and $\phi_2$ is used for the $q_{\phi_2}(m|x)$ part. We assume in $q$ that given $X$, $Y$ does not contain further information about $M$. As a consequence, during inference, $Y$ and $M$ are conditionally independent given $X$, although it is not implied in the $p$ graphical model (i.e., the causal model). Therefore in $q_{\phi_2}(m|x)$ we only extract the information of $M$ from $X$, which, as we show later, allows deep CAMA to learn unseen manipulations.

The network architecture is presented in Figure 4. For the $p$ model, the cause variables $Y$, $Z$ and $M$ are first transformed into feature vectors $h_Y, h_Z$ and $h_M$. Later, these features are merged together and then passed through another neural network to produce the distributional parameters of $p_\theta(x|y, z, m)$. For the approximate posterior $q$, two different networks are used to compute the distributional parameters of $q_{\phi_2}(m|x)$ and $q_{\phi_1}(z|x, y, m)$, respectively.

**Model training** We describe the training procedure for two different scenarios. First, assume that during training, the model observes clean data $\mathcal{D} = \{(x_n, y_n)\}$ only. In this case we set the manipulation variable $M$ to a null value, e.g. $do(m = 0)$. and train deep CAMA by maximizing the likelihood function $\log p(x, y | do(m = 0))$ under training data. As there is no incoming edges to the manipulation variable $M$, the do-calculus can be reduced to the conditional distribution $p(x, y | do(m = 0)) = p(x, y | m = 0)$. Since this marginal distribution is intractable, we instead maximize the intervention evidence lower-bound (ELBO) with $do(m = 0)$, i.e. $\max_{\theta, \phi} \mathbb{E}_{\mathcal{D}}[\text{ELBO}(x, y, do(m = 0))]$, with the ELBO defined as (derived in Appenfix **??**):

$$\text{ELBO}(x, y, do(m = 0)) := \mathbb{E}_{q_{\phi_1}(z | x, y, m = 0)} \left[ \log \frac{p_\theta(x | y, z, m = 0) p(y) p(z)}{q_{\phi_1}(z | x, y, m = 0)} \right]. \tag{3}$$

If manipulated data $\mathcal{D}'$ is available during training, then similar to data augmentation and adversarial training [17, 48, 30], we can augment the training data with such data. We still use the intervention ELBO (3) for clean data. For the manipulated instances, we can either use the intervention ELBO with $do(m = m_0)$ when the manipulated data $\mathcal{D}' = \{(m_0(x), y)\}$ is generated by a known intervention $m_0$, or, as done in our experiments, infer the latent variable $M$ for unknown manipulations. This is achieved by maximizing the ELBO on the joint distribution $\log p(x, y)$ using manipulated data:

$$\text{ELBO}(x, y) := \mathbb{E}_{q_\phi(z, m | x, y)} \left[ \log \frac{p_\theta(x, y, z, m)}{q_\phi(z, m | x, y)} \right], \tag{4}$$

so the total loss function to be maximized is defined as

$$\mathcal{L}_{\text{aug}}(\theta, \phi) = \lambda \mathbb{E}_{\mathcal{D}}[\text{ELBO}(x, y, do(m = 0))] + (1 - \lambda) \mathbb{E}_{\mathcal{D}'}[\text{ELBO}(x, y)]. \tag{5}$$

This training procedure only requires knowledge on whether the training data is clean (in such case we set $m = 0$) or manipulated (potentially with unknown manipulation). In the manipulated case the model does not require explicit label for $M$ and performs inference on it instead. For test data, only the input X is given, and the labels for both $Y$ and $M$ are not available.

Our causally consistent model effectively disentangles the latent representation: $Z$ models the unknown causes in the clean data, such as personal writing style; and $M$ models possible manipulations or interventions on the underlying factors, which the model should be robust to, such as shift, rotation, noise etc. From a causal perspective, the mechanism of generating $X$ from its causes is invariant to the interventions on $M$. Thus, in our model the functional relationships $Y \to X$ and $Z \to X$ remain intact even in the presence of manipulated data. As a result, deep CAMA's can still generalize to unseen manipulations even after seeing lots of manipulated datapoints from other manipulations, in contrast to the behavior of discriminative DNNs as shown in Figure 1.

**Prediction** We wish our model to be robust to an unseen intervention on test data $\tilde{\mathcal{D}}$, i.e. $M$ is unknown at test-time. Here deep CAMA classifies an unseen test data $x^*$, using a Monte Carlo approximation to Bayes' rule with samples $m^u \sim q_{\phi_2}(m | x)$, $z_c^k \sim q_{\phi_1}(z | x^*, y_c, m^u)$:

$$p(y^* | x^*) = \frac{p(x^* | y^*) p(y^*)}{p(x^*)} \approx \text{softmax}_{c=1}^C \left[ \log \sum_{k=1}^K \frac{p_\theta(x | y, z_c^k, m^u) p(y_c) p(z)}{q_{\phi_1}(z_c^k | x^*, y_c, m^u)} \right]. \tag{6}$$

In experiments we use 1 sample of $m \sim q(m | x)$ and $K$ samples of $z \sim q(z | x, y, m)$ associated with each $m$ sample. Given the sampled $m$ and $z$ instances, we can compute the log-ratio term for each $y = c$ (as an approximation to $\log p(x, y = c)$), and apply softmax to compute Bayes' rule and obtain the predictive distribution.

**Test-time fine-tuning** Deep CAMA can also be adapted to the unseen manipulations presented in test data *without labels*. From the causal graph, the conditional distributions $p(x | y)$ and $p(x | z)$ are invariant to the interventions on $X$ based on the modularity property. However, we would like to learn the manipulation mechanism $M \to X$, and, given that the number of possible interventions on $M$ might be infinity, the model may be underfitted for this functional relationship, given limited data. In this regard, fine-tuning on the current observation can be beneficial, thereby hopefully making deep CAMA more robust. As shown in Figure 4, for the generative model, we only fine-tune the networks that are dependent only on $M$, i.e. $\text{NN}_M^p$ by maximizing the ELBO of $\log p(x)$:

$$\text{ELBO}(x) := \log \left[ \sum_{c=1}^C \exp[\text{ELBO}(x, y_c)] \right]. \tag{7}$$

To reduce the possibly negative effect of fine-tuning to model generalization, we use a shallow network for $\text{NN}^p_{merge}$ and deep networks for $\text{NN}^p_M$, $\text{NN}^p_Y$ and $\text{NN}^p_Z$. We also fine-tune the network $\text{NN}^q_M$ for the approximate posterior $q$ since $M$ is involved in the inference of $Z$. In sum, in fine-tuning the selective part of the deep CAMA model is trained to maximize the following objective:

$$\mathcal{L}_{\text{ft}}(\theta, \phi) = \alpha\mathbb{E}_\mathcal{D}[\text{ELBO}(x, y)] + (1 - \alpha)\mathbb{E}_{\tilde{\mathcal{D}}}[\text{ELBO}(x)]. \tag{8}$$

Note that the intervention ELBO can also be used for $\mathcal{D}$, in which we explore such option in some of the experiments. Importantly, there may exist infinitely many manipulations and it is impossible to train with all of them together. So by fine-tuning in a just-in-time manner, the model can be adapted to unseen manipulation at test time, which is confirmed in our experiments.

The time complexity of CAMA training is in the same order of training a regular variational auto-encoder [24, 39]. For predictions, test-time fine-tuning requires a small amount of additional time, as only a small fraction of data is needed for fine-tuning, see Figure 8 and Figure **??** in appendix.

## 3.2 Deep CAMA for generic measurement data

We now discuss a generic version of deep CAMA to handle multimodality in measurement data. To predict the target variable $Y$ in a directed acyclic graph, only variables in the Markov blanket of $Y$ (shown in Figure 5) are needed. This includes the parents ($A$), children ($X$), and co-parents ($C$) of the target $Y$. Similar to the single modal case above, here a valid manipulation can only be independent mechanisms applied to $X$ or $C$ to ensure that both $Y$ and the relationship from $Y$ to $X$ remain intact.

We design the generic deep CAMA (shown in Figure 6) following the causal process in Figure 5. Unlike discriminative DNNs where $A$, $C$ and $X$ are used together to predict $Y$ directly, we consider the full causal process and treat them separately. Building on the deep CAMA for single modality data, we add the extra consideration of the parent and observed co-parent of $Y$, while modelling the latent unobserved cause in $Z$ and potential manipulations in $M$. We do not need to model manipulation on $C$ as they are out of the Markov Blanket of $Y$. Thus, our model and the approximate inference network are defined as

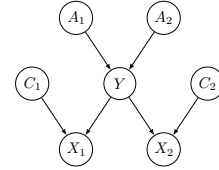

Figure 5: The Markov Blanket of target variable $Y$

$$p_\theta(x, y, z, m, a, c) = p(a)p(m)p(z)p(c)p_{\theta_1}(y|a)p_{\theta_2}(x|y, c, z, m), \quad (9)$$

$$q_\phi(z, m|x, y, a, c) = q_{\phi_1}(z|x, y, m, a, c)q_{\phi_2}(m|x). \tag{10}$$

Training, fine-tuning and prediction proceed in the same way as in the single modality case (Section 3.1) with $do(m)$ operations and Monte Carlo approximations. As we only fine-tune the networks that are dependent on $M$, similar reasoning indicates that the multimodality deep CAMA is robust to manipulations directly on the effect variable $X$.

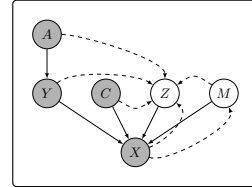

Figure 6: Graphical presentation of deep CAMA for generic measurement data.

Our model is also robust to changes of $X$ caused by intervention on the co-parents $C$ by design. By our definition of valid manipulation, perturbing $C$ is valid as it only leads to the changes in $X$. If the underlying model from $Y$ and $C$ to $X$ remains the same, and the trained model learns $p(x|y, c)$ *perfectly*, then our model is perfectly robust to such changes, due to Bayes' rule for prediction:

$$p(y|a, x, c) = \frac{p(y|a)p(a)p(c)p(x|y, c)}{p(a)p(c)\int_y p(y|a)p(x|y, c)} = \frac{p(y|a)p(x|y, c)}{\int_y p(y|a)p(x|y, c)}, \tag{11}$$

and the manipulations on $C$ (thus changing $X$) do not affect the conditional distribution $p(x|y, c)$ in the generative classifier (Eq. 11). In contrast, discriminative DNNs concatenate $X$, $C$, $A$ together and map these variables to $Y$, and therefore it fails to make use of the invariant mechanisms.

**Causal consistency in model design** To build the deep CAMA model for measurement data (Figure 6), it requires a causally consistent specification of $C$, $X$, and $A$ variables in the graphical model. Thus, the causal view is crucial not only for valid manipulation definitions but also for model design. In this work, we assume that the causal relationship of the observed variables is given and use it to build the model, although in experiments we also empirically investigate the cases when this assumption is violated. This is different from the line of causality research aiming at finding causal relations from observational data, where suitable assumptions are always needed [45, 54].

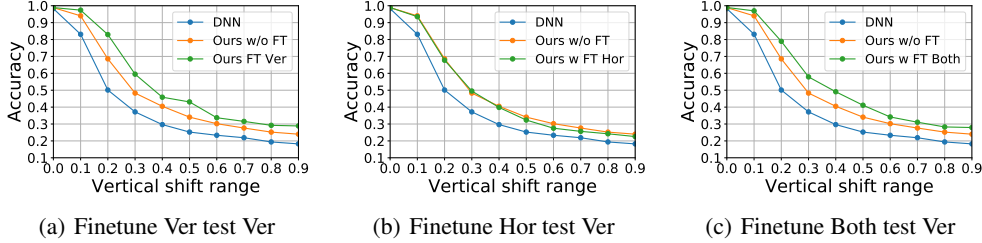

(a) Finetune Ver test Ver      (b) Finetune Hor test Ver      (c) Finetune Both test Ver

Figure 7: Model robustness results on vertical shifts. (See Appendix **??** for results on horizontal shifts)

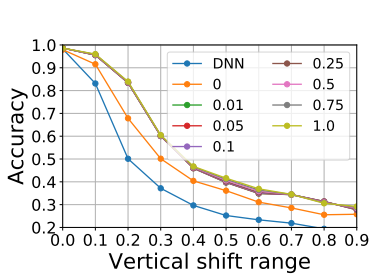

Figure 8: Performance regarding different percentages of test data used for fine-tuning manipulation

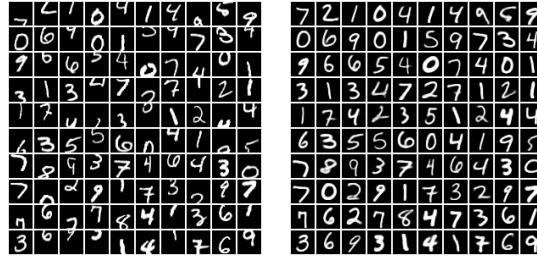

(a) Vertically shifted data    (b) do(m=0) with $z$ & $y$ from the vertical shifted data

Figure 9: Visualization of the disentangled representation.

Orthogonal to our work, there are many methods for causal discovery (see, e.g., [45, 14]), for both observational data and interventional data, and remains an active research direction.

# 4 Experiments

We evaluate the robustness of deep CAMA for image classification using both MNIST and a binary classification task derived from CIFAR-10 (Appendix **??**). Furthermore, we demonstrate the behaviour of our generic deep CAMA for measurement data. We evaluate the performance of CAMA on both manipulations and adverserial examples generated using the CleverHans package [33].

## 4.1 Robustness test on image classification with Deep CAMA

We first demonstrate the robustness of our model to vertical (Ver) and horizontal (Hor) shifts.

**Training with clean MNIST** Figure 7 shows the robustness results on vertical shifts for deep CAMA trained on clean data only.[2] Deep CAMA without fine-tuning (orange lines) perform slightly better than a DNN (blue lines). The advantage of deep CAMA is clear when fine-tuning is used at test time (green lines): fine-tuning on manipulated test data with the same shift clearly improves the robustness of the network (Figure 7(a)). Furthermore Figure 7(b) shows that deep CAMA generalizes to *unseen* vertical shifts after fine-tuning with horizontal shifts. Lastly, Figure 7(c) shows the improved robustness of our model when both types of manipulation are used for fine-tuning. All these results indicate that our model is capable of learning manipulations in an unsupervised manner, without deteriorating the generalization ability to unseen manipulations. Comparisons to other deep generative model baselines are presented in the appendix **??**, and the results show the advantage of CAMA (especially with fine-tuning).

**Training with augmented MNIST** We explore the setting where the training data is augmented with manipulated data. As discussed in Section 3.1, here deep CAMA naturally learns disentangled representation due to its causal reasoning capacity. Indeed this is confirmed by Figure 9, where panel 9(b) shows the reconstructions of manipulated data from panel 9(a) with $do(m = 0)$. In this case the model keeps the identity of the digits but moves them to the center of the image. Recall that $do(m = 0)$ corresponds to clean data which contains centered digits. This shows that deep CAMA can disentangle the intrinsic unknown style $Z$ and the shifting manipulation variable $M$.

We show the robustness results of deep CAMA with data augmentation (shift range 0.5) in Figure 10. A comparison to results in Figures 1 clearly shows the advantage of deep CAMA over disciminative DNNs: in addition to *seen* perturbations in augmented data, deep CAMA is also robust to *unseen* manipulation. Take the vertical shift test in panel 10(a) for example. When vertically shifted data are augmented to the training set, the test performance without fine-tuning (green line) is significantly bet-

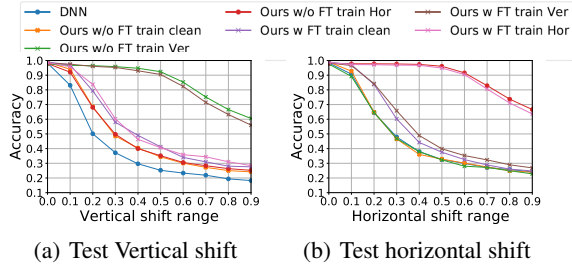

(a) Test Vertical shift     (b) Test horizontal shift

Figure 10: Augmented MNIST robustness results.

ter. Further, fine-tuning (brown line) brings in even larger improvement for large scale shifts. On the other hand, deep CAMA maintains robustness on vertically shifted data when trained with horizontally shifted data. By contrast, training discriminative DNNs with one manipulation might even hurt its robustness to unseen manipulations (Figure 1). Therefore, our model does not overfit to a specific type of manipulations, at the same time further fine-tuning can improve the robustness against new manipulations (pink line). The same conclusion holds in panel 10(b).

We also quantify the amount of manipulated data required for fine-tuning in order to improve the robustness of deep CAMA models. As shown in Figure 8, even using $1\%$ of the manipulated data is sufficient to learn the vertical shift manipulation presented in the test set.

**Adversarial robustness on MNIST**    We further test deep CAMA's robustness to two adversarial attacks: fast gradient sign method (FGSM) [16] and projected gradient descent (PGD) [30]. Note that these attacks are specially developed for images with the small perturbation constraint. They are not guaranteed to be valid attacks by our definition, as the manipulation depends on $Y$, which has the risk of changing the ground-truth class label. Such risk has also been discussed in Elsayed et al. [10].

Figure 11 show the results on models trained on clean images only. Deep CAMA is significantly more robust to both attacks than the DNN, and with fine-tuning, deep CAMA shows additional $20\% - 40\%$ accuracy increase. The clean data test accuracy after fine-tuning remains the same thanks to the causal consistent model design. Comparisons to more baselines can be found in appendix **??**.

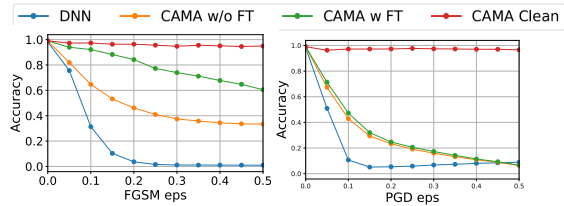

Figure 11: Test accuracy on MNIST adversarial examples.

In Appendix **??** we also perform adversarial robustness tests on a natural image binary classification task derived from CIFAR-10. Again deep CAMA out-performs a discriminative CNN even without fine-tuning; also fine-tuning provides additional advantages without deteriorating the clean accuracy.

## 4.2   Robustness test on measurement based data with generalized Deep CAMA

Our causal view on valid manipulations allows us to test model robustness on generic measurement data. Since there is no real-world dataset with known underlying causal graph, we generate synthetic data following a causal process with non-linear causal relationships (see Appendix **??**), and perform robustness tests therein. We use Gaussian variables for $A$, $C$ and $X$, and categorical variables for $Y$.

**Shifting tests**    We shift selected variables up- or down-scale, which resembles a type of noise in real-world data: different standards on subjective quantities, such as pain scale in diagnosis. We present the up-scale results below; the down-scale results in Appendix **??** are of similar behaviour.

The first test shifts the co-parents $C$ while keeping the relationship $C \rightarrow X$ static,

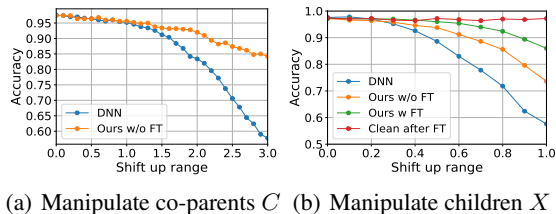

(a) Manipulate co-parents $C$   (b) Manipulate children $X$

Figure 12: Shift-up robustness results on measurement data.

resulting in a corresponding change in $X$. Figure 12(a) shows the result: deep CAMA is significantly more robust. In particular, with increasing shift distortions, the accuracy of the DNN drops drastically. This corroborates our theory in Section 3.2 that manipulations in $C$ has little impact on deep CAMA's prediction.

The second test shifts $X$ only, and the model only uses clean data for training. From Figure 12(b), deep CAMA is again much more robust even without fine-tuning (orange vs blue). This robustness is further improved by fine-tuning (in green) without negatively affecting clean test accuracies (in red). This confirms that fine-tuning learns the influence of $M$ without affecting the causal relationships between $Y$ and $Z$.

**Adversarial robustness**   In this test we only allow attacks on the children $X$ and co-parents $C$ according to our definition of valid attacks. This applies to all the models in test. Figure 13 shows again that deep CAMA are significantly more robust to adversarial attacks, and fine-tuning further improves robustness while keeping high accuracy on clean test examples.

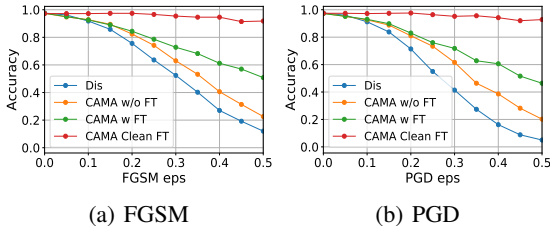

(a) FGSM          (b) PGD

Figure 13: Adv. robustness results on measurement data.

**Violated assumptions**   So far we assume that the causal relationship among variables of interests are provided, either by domain experts or by running causal discovery algorithms. However, for both cases, there exists possibility that the provided causal graph is not perfect. We thus test the case where parts of deep CAMA's graphical model are mis-specified. It may happen when we have a wrong understanding of the data generating process, or when a causal discovery algorithm fail for multiple possi-

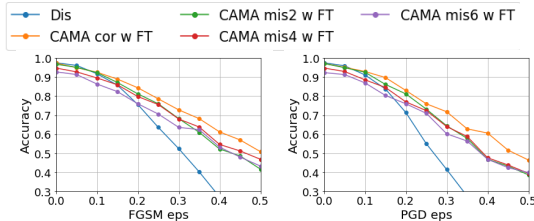

Figure 14: Test accuracy on measurement data for CAMA with mis-specified model ("mis2" denotes that 2 children nodes are mis-specified as co-parents).

ble reasons. We use the same synthetic data as before, but to simulate this mis-specification setting, we intentionally use some children nodes in the ground truth causal graph as the co-parent nodes in deep CAMA's graphical model. Indeed this has negative impact on performance as shown in Figure 14, however, deep CAMA remains to be more robust than the discriminative DNN when the mis-specification is not too severe. This again demonstrates the importance of causal consistency in model design. Additional results can be found in Appendix **??**. It also shows the importance of working closely with domain experts as well as careful evaluations of existing causal discovery algorithms [49, 14].

## 5   Related Work

**Adversarial robustness**   Adversarial attacks can easily fool a discriminative DNN by adding imperceptible perturbations [7, 2, 6, 47, 32]. Adversarial training [30, 48] has shown some success in defending attacks; however, it requires knowledge of the adversary to present the perturbation to the model during training. Even so, a discriminative model after adversarial training is vulnerable to unseen manipulations. Meanwhile, existing theoretical works [9, 52, 11] evaluate the robustness of a classifier trained on clean data and show there is no free lunch against attacks. However, such study does not necessary apply to our proposed model with test-time fine-tuning. Deep generative modelling has been applied as a defence mechanism to adversarial attacks. One line of work considered denoising adversarial examples before feeding these inputs to the discriminative classifier [44, 41]. Another line of research revisited (deep) generative classifiers and provided evidence that they are more robust to adversarial attacks [29, 43, 26]. Lastly, the Monte Carlo estimation techniques are also used in Bayesian neural networks (BNNs) which have also been shown to be more robust than their deterministic counterparts [27, 12]. However, Li et al. [29] shows that the advantage of generative classifiers over BNNs is due to the difference of the generative/discriminative nature, rather than the

usage of Monte Carlo estimates. Deep CAMA belongs to the class of generative classifiers, on top of Li et al. [29] we further demonstrates its improved robustness to unseen manipulations.

**Causal learning**  Causal inference has a long history in statistical research [45, 35, 38, 36], but to date, the causal view has not been widely incorporated to robust prediction under unseen manipulations. The most relevant work is in the field of transfer learning and domain adaption, where the difference in various domains are treated as either target shift or conditional shift from a causal perspective [53, 46, 56, 15, 21]. Extensions of the domain adaptation work also discuss robust predictors across different domains [40, 19, 3], in which the domain is specified either explicitly or though exemplar paired points. For example, invariant risk minimization (IRM) [3], focusing on supervised learning with single modality data, proposes a regularizer in the loss function to encourage invariant predictions across different environments with given environments label in the training set. To make it adaptive and automated, domain adaptation has been viewed as problem of inference on graphical models, which provide a compact representation of the changeability of the data distribution and can be directly learned from data [55]. By contrast, our proposed method considers *unseen* manipulations without relying on information of domain shifts. In addition, we address a more general problem – interventions on a datapoint – whereas in domain adaption interventions are considered on a dataset level. Another related area is causal feature selection [1], where causal discovery is applied first, and then features in the Markov Blanket of the prediction target are selected. We also note that CAMA's design is aligned with causal and anti-causal learning analyses [42, 22], in that CAMA models the causal mechanism $Y \rightarrow X$ and use Bayes' rule for anti-causal prediction. On the differences, CAMA is not limited to only two endogenous variables; rather it provides a generic design to handle latent causes that correspond to both intrinsic variations and data manipulations.

**Disentangled representations**  Learning disentangled representations has become a trendy research topic in recent representation learning literature. Considerable effort went to developing training objectives, e.g. $\beta$-VAE [20] and other information theoretic approaches [23, 8]. Additionally, different factorization structure in graphical model design has also been explored for disentanglement [31, 28]. The deep CAMA model is motivated by the causal process of data manipulations, which differs from the model used in Narayanaswamy et al. [31] in that the latent variables have different meanings. This difference is elaborated in Appendix **??**. Furthermore test-time fine-tuning allows deep CAMA to better adapt to unseen manipulations, which is shown to be useful for improving robustness.

## 6   Conclusion and Discussion

We provided a causal view on the robustness of neural networks, showing that the vulnerability of discriminative DNNs can be explained by the lack of causal reasoning. We defined valid attacks under this causal view, which are interventions of data though the causal factors which are not the target label or the ancestor of the target label. We further proposed a deep causal manipulation augmented model (deep CAMA), which follows the causal relationship in the model design, and can be adapted to unseen manipulations at test time. Our model has demonstrated improved robustness, even without adversarial training. When manipulated data are available, our model's robustness increases for both seen and unseen manipulation.

The ground truth causal graph is often complicated, but the CAMA graphical model simplifies it by grouping causes into different types, and treating each type as a single high dimensional variable. This is sufficient for our application at hand, but more fine-grind causal model may be needed for others. For example, one might prefer learning different type of manipulation separately, which may require working with a challenging causal graph. We leave the investigation in future work.

Our framework is generic, however, manipulations can change over time, and a robust model should adapt to these perturbations in a continuous manner. Our framework thus should be adapted to online learning or continual learning settings. In future work, we will explore the continual learning setting of deep CAMA where new manipulations come in a sequence. In addition, our method is designed for generic class-independent manipulations, and therefore a natural extension would consider class-dependent manipulations where $M$ is an effect of $Y$ or there is a confounder for $M$ and $Y$. Lastly, our design excludes gradient-based adversarial attacks which is dependent on both the target and the victim model. As such attacks are commonly adopted in machine learning, we would also like to extend our model to such scenarios.

## Acknowledgement

KZ would like to acknowledge the support by the United States Air Force under Contract No. FA8650-17-C-7715. CZ and YL would like to acknowledge Nathan Jones for his support with computing infrastructure; Tom Ellis and Luke Harries for feedback regarding the manuscript.

## Broader Impact

In this work, we provide a causal perspective on the robustness of deep learning, and propose a causal consistent deep generative model as an instance to improve the robustness of model regarding unseen manipulations. We view the robustness of AI solution under unseen manipulation as a key factor for many AI-aided decision making system to be trusted. While we do not intent to claim we have solved this problem perfectly (especially concerning large-scale and real-life applications), our work has shown great improvement over existing methods regarding the robustness towards unseen manipulations. We hope our research can inspire more solutions towards the final goal of AI safety.

## Footnotes

[2]Results on horizontal shifts are presented in Appendix **??**, and the conclusions there are similar.

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
