[Supplementary Material]

# A Derivation Details

## A.1 The intervention ELBO

When training with clean data $\mathcal{D} = \{(x_n, y_n)\}$, we set the manipulation variable $M$ to a null value, e.g. $do(m = 0)$. In this case we would like to maximise the log-likelihood of the *intervened* model, i.e.

$$\max_\theta \mathbb{E}_\mathcal{D}[\log p_\theta(x, y | do(m = 0))].$$

This log-likelihood of the intervened model is defined by integrating out the unobserved latent variable $Z$ in the intervened joint distribution, and from do-calculus we have

$$\log p_\theta(x, y | do(m = 0)) = \log \int p_\theta(x, y, z | do(m = 0)) dz$$
$$= \log \int p_\theta(x | y, z, m = 0) p(y) p(z) dz. \tag{12}$$

A variational lower-bound (or ELBO) of the log-likelihood uses a variational distribution $q(z|\cdot)$

$$\log p_\theta(x, y | do(m = 0)) = \log \int p_\theta(x | y, z, m = 0) p(y) p(z) \frac{q(z|\cdot)}{q(z|\cdot)} dz$$
$$\geq \mathbb{E}_{q(z|\cdot)} \left[ \log \frac{p_\theta(x | y, z, m = 0) p(y) p(z)}{q(z|\cdot)} \right]. \tag{13}$$

The lower-bound holds for arbitrary $q(z|\cdot)$ as long as it is absolutely continuous w.r.t. the posterior distribution $p_\theta(z | x, y, do(m = 0))$ of the intervened model. Now recall the design of the inference network/variational distribution in the main text:

$$q_\phi(z, m | x, y) = q_{\phi_1}(z | x, y, m) q_{\phi_2}(m | x),$$

where $\phi_1$ and $\phi_2$ are the inference network parameters of the corresponding variational distributions. Performing an intervention $do(m = 0)$ on this $q$ distribution gives

$$q_\phi(z | x, y, do(m = 0)) = q_{\phi_1}(z | x, y, m = 0).$$

Defining $q(z|\cdot) = q_{\phi_1}(z | x, y, do(m = 0))$ and plugging-in it to eq. (13) return the *intervention* ELBO objective (3) presented in the main text.

## A.2 The ELBO for unlabelled test data

The proposed fine-tuning method in the main text require optimising the marginal log-likelihood $\log p_\theta(x)$ for $x \sim \tilde{\mathcal{D}}$, which is clearly intractable. Instead of using a variational distribution for the unobserved class label $Y$, we consider the variational lower-bound of $\log p_\theta(x, y)$ for all possible $y = y_c$:

$$\log p_\theta(x, y) = \log \int p_\theta(x, y, z, m) dz dm$$
$$= \log \int p_\theta(x, y, z, m) \frac{q_\phi(z, m | x, y)}{q_\phi(z, m | x, y)} dz dm \tag{14}$$
$$\geq \mathbb{E}_{q_\phi(z, m | x, y)} \left[ \log \frac{p_\theta(x, y, z, m)}{q_\phi(z, m | x, y)} \right] := \text{ELBO}(x, y).$$

Since both logarithm and exponent functions preserve monotonicity, and for all $y_c, c = 1, ..., C$ we have $\log p_\theta(x, y_c) \geq \text{ELBO}(x, y_c)$, we have

$$\log p_\theta(x, y_c) \geq \text{ELBO}(x, y_c), \forall c \implies p_\theta(x, y_c) \geq \exp[\text{ELBO}(x, y_c)], \forall c$$

$$\implies \log p(x) = \log \left[ \sum_{c=1}^C p_\theta(x, y_c) \right] \geq \log \left[ \sum_{c=1}^C \exp[\text{ELBO}(x, y_c)] \right] := \text{ELBO}(x),$$

which justifies the ELBO objective (7) defined in the main text.

# B    Addition discussions on comparisons to Narayanaswamy et al. [32]

Narayanaswamy et al. [32] proposed a semi-supervised learning algorithm to learn disentangled representation for computer vision tasks. Their approach extends VAE with two latent variables $Y$ and $Z$ to model images $X$. They provide partially observed $Y$ which are "interpretable" depending on the computer vision application context. They achieve meaningful synthetic image generation by sample different latent variables in $Y$ and $Z$.

On the other hand, the proposed deep CAMA model focuses on a causal view for robustness of neural networks to unseen manipulations. Firstly, the latent variables have very different meanings. Apart from the fact that $Y$ is solely used to represent the prediction target, the $M$ and $Z$ variables are designed to separate the latent factors that can or cannot be *artificially* intervened by the adversary. Secondly, the fine-tuning algorithm provides a test-time adaptation scheme for the deep CAMA model (thus enabling adaptation to any unseen manipulations), which is different from Narayanaswamy et al. [32] where the trained model is directly applied in the testing time. Importantly, the fine-tuning method updates the model parameter in a selective manner, which is motivated by our analysis on the causal generation process of noisy data. It also allow the model to generate to unseen manipulation in test time.

Another important contribution of the paper is the generalization of deep CAMA to generic measurement data. In this case the causal graph of the data generation process plays a key role in model design, since now deep CAMA also models all the variables in the Markov blanket of $Y$, which is clearly beyond the factor model considered in Narayanaswamy et al. [32]. Our empirical study on the causal consistency of model design (see the "violated assumptions" experiments) clearly shows that a consistent design of the model to the underlying causal graph is key to both the robustness of the model and the efficiency of fine-tuning for unseen manipulations.

# C    Additional Results

## C.1    Additional results of DNN behavior

**CNN**    We also performed experiments using different DNN network architectures. The convolution layers in CNN are designed to be robust to shifts. Thus, we test these vertical and horizontal shifts with a standard CNN architecture as used in `https://keras.io/examples/cifar10_cnn/`. 4 convolution layers are used in this architecture.

Figure 15 shows the performance against different shifts. We see that adding vertical shifts to the training data clearly harmed the robustness performances to unseen horizontal shifts as shown in 16(b). Adding horizontal shifted images in training did not influences the performance on vertical shifts much. Thus, we see that using different architectures of DNN, even the one that are designed to be robust to these manipulations, lack of generalization ability to unseen data is a common problem.

(a) Test Vertical shift

(b) Test Horizontal shift

Figure 15: Robustness results for DNNs against different manipulations on MNIST using CNN. Panels (a) and (b) show the accuracy on classifying noisy test data generated by shifting the digits vertically (vt) and horizontally (ht). It shows that data augmentation during training makes generalization to unseen shifts worse (orange versus blue lines).

(a) Test Vertical shift                    (b) Test Horizontal shift

Figure 16: Robustness results for DNNs against different manipulations on MNIST using a large MLP. Panels (a) and (b) show the accuracy on classifying noisy test data generated by shifting the digits vertically (vt) and horizontally (ht). It shows that data augmentation during training makes generalization to unseen shifts worse (orange versus blue lines).

**Enlarge Network Size**    Here we exam whether network capacity has any influence on the robustness performance to unseen manipulation. We use a wider network with [1024, 512, 512, 1024] units in each hidden layer instead of [512, 256, 126, 512] sized network in the paper. Figure 16 shows the robustness performance using this enlarged network. We observe the similar degree of over-fitting to the augmented data. The penalization ability shows no improvement by enlarging the network sizes.

**ZCA Whitening Manipulation**    Our result does not limited to shifts, it generalizes to other manipulations. Figure 17 compare the result from training with clean images and training with ZCA whitening images added. We see that adding ZCA whitening images in training harm both robustness against vertical shift and horizontal shift.

(a) Test Vertical shift                    (b) Test Horizontal shift

Figure 17: ZCA Whitening manipulation result. Figure shows the robustness results for DNNs against different manipulations on MNIST using CNN. The blue curve shows that result from training with clean data. The orange curve shows that result from training with zca whitening data added.

## C.2   Additional Results with MNIST experiment

We present the test result regarding the horizontal shifts in Figure 18 which is under the same setting as Figure 7 . The results support the same conclusion: fine-tuning on data with different manipulation does not decrease the generalization ability with our model, which is shown in Figure 18(a) where the green line (fine-tuning with vertical shifts) and the orange line (without fine-tuning) overlaps; Fine-tuning in testing time with the test data without label significantly improves the performance which is shown in Figure 18(b) and 18(c) comparing to the orange line and the green line. Note that we have no intention to claim superior robustness of generative models to unseen manipulations without fine-tuning. Instead, we would like to show that they are able to obtain competitive performance when compared with discriminative models (gren vs blue lines). Our observation is consistent with previous work that when robustness is concerned, generative models are at least as competitive as discriminative ones [30].

(a) Finetune VT test HT    (b) Finetune HT test HT    (c) Finetune Both test HT

Figure 18: Model robustness results on horizontal shifts

Figure 19: Performance regarding different percentage of test data used for fine-tuning manipulation of horizontal shift without using $do(m) = 0$ for the cleaning training data during fine-tuning.

Figure 20: Performance regarding different percentage of test data used for fine-tuning manipulation of vertical shift using $do(m) = 0$ for the cleaning training data during fine-tuning.

In addition to Figure 8, We also show the result testing with Vertical shift show in Figure 19, where a smaller $N_M^p$ network ([dimM, 500, 500]) is used. The conclusion is the same was using the vertical shift. We need very few data for fine-tune. More than $1\%$ data is sufficient.

Similar as Figure 8, we show the result using different percentage of data for fine-tuning in this experiment setting in 20.

### C.3   Adversarial attack test on natural image classification

We evaluate the adversarial robustness of deep CAMA when trained on natural images. In this case we follow Li et al. [30] and consider *CIFAR-binary*, a binary classification dataset containing "airplane" and "frog" images from CIFAR-10. We choose to work with CIFAR-binary because VAE-based fully generative classifiers are less satisfactory for classifying clean CIFAR-10 images ($< 50\%$ clean test accuracy). The deep CAMA model trained with data augmentation (adding Gaussian noise with standard deviation 0.1, see objective (5)) achieves $88.85\%$ clean test accuracy on CIFAR-binary, which is on par with the results reported in Li et al. [30]. For reference, a discriminative CNN

(a) FGSM    (b) PGD

Figure 21: Adversarial robustness results on CIFAR-binary.

Figure 22: Comparisons to classifiers based on other deep latent variable models, including CVAE [45] and DVIB [1]. This again shows the benefit of the generative model structure in CAMA.

with $2\times$ more channels achieves $95.60\%$ clean test accuracy. Similar to previous sections we apply FGSM and PGD attacks with different $\epsilon$ values to both deep CAMA and the discriminative CNN, and evaluate classification accuracies on the adversarial examples before and after finetuning.

Results are reported in Figure 21. For both FGSM and PGD tests, we see that deep CAMA, before finetuning, is significantly more robust to adversarial attacks when compared with a discriminative CNN model. Regarding finetuning, although PGD with large distortion ($\epsilon = 0.2$) also fools the finetuning mechanism, in other cases finetuning still provides modest improvements ($5\%$ to $8\%$ when compared with the vanilla deep CAMA model) without deteriorating test accuracy on clean data. Combined with adversarial robustness results on MNIST, we conjecture that with a better generative model on natural images the robustness of deep CAMA can be further improved.

## C.4 Additional Baselines

we present in Figure 22 the robustness of CVAE [45] & deep variational information bottleneck (DVIB) [1] to vertical shifts and PGD attacks. Both baselines use *discriminative* latent variable models for classification, and they perform either close to CAMA in vertical shift tests, or much worse than CAMA in PGD attack tests. This clearly shows the advantage of CAMA as a generative classifier, especially with fine-tuning. Similar results have been reported in Li et al. [30], and our work provides extra advantages due to the use of a causally consistent model and the fine-tuning method motivated by causal reasoning.

## C.5 Additional results with measurement data

**Shifting tests**   Figure 23 demonstrates the results on shifting the selected variables down-scale. The result is consistent with the shifting up-scale scenario in the main text. Deep CAMA is significantly more robust than baseline methods in both cases of manipulating co-parents and manipulating children.

**Additional Experiments regarding assumption violation**   We consider a more challenging scenario of a even more serious violation of the causal consistency assumption. In this case deep CAMA's graphical model is mis-specified by swapping the variables in the children/co-parent positions of the ground truth causal graph. Thus, in this case, the "mis1" experimental setting means two nodes are in the wrong positions (compared to the main text experiments where only one node is specified in the wrong position). Figure 24 shows the result of this co-parents/children swapping experiment. When only one pair of the variables is swapped (mis1, orange), the performance is still competitive with the causal consistent version of CAMA (cor). However, as we increase the number of swapped nodes, the robustness performance becomes significantly worse. This is again shows the importance of causal consistency for model design; still deep CAMA remains to be reasonably robust in the case of minor mis-specifications.

(a) Manipulate co-parents $C$.

(b) Manipulate children $X$.

Figure 23: Shift Downscale

Figure 24: Test accuracy on measurement when causal relationship assumptions are violated. Swapping experiment.

# D Experimental settings

**Network architecture**

- MNIST experiments:
  - Discriminative DNN: The discriminate model used in the paper contains 4 densely connected hidden layer of $[512, 256, 126, 512]$ width for each layer. ReLU activations and dropout are used with dropout rate $[0.25, 0.25, 0.25, 0.5]$ for each layer.
  - Deep CAMA's $p$ networks: we use $\dim(Y) = 10, \dim(Z) = 64$ and $\dim(M) = 32$.
    $\text{NN}_Y^p$: an MLP of layer sizes $[\dim(Y), 500, 500]$ and ReLU activations.
    $\text{NN}_Z^p$: an MLP of layer sizes $[\dim(Z), 500, 500]$ and ReLU activations.
    $\text{NN}_M^p$: an MLP of layer sizes $[\dim(M), 500, 500, 500, 500]$ and ReLU activations.
    $\text{NN}_{\text{merge}}^p$: an projection layer which projects the feature outputs from the previous networks to a 3D tensor of shape $(4, 4, 64)$, followed by 3 deconvolutional layers with stride 2, SAME padding, filter size $(3, 3, 64, 64)$ except for the last layer $(3, 3, 64, 1)$. All the layers use ReLU activations except for the last layer, which uses sigmoid activation.
  - Deep CAMA's $q$ networks:
    $\text{NN}_M^q$: it starts from a convolutional neural network (CNN) with 3 blocks of $\{\text{conv}3 \times 3, \text{max-pool}\}$ layers with output channel size 64, stride 1 and SAME padding, then performs a reshape-to-vector operation and transforms this vector with an MLP of layer sizes $[4 \times 4 \times 64, 500, \dim(M) \times 2]$ to generate the mean and log-variance of $q(m|x)$. All the layers use ReLU activation except for the last layer, which uses linear activation.

$\mathrm{NN}_Z^q$: first it uses a CNN with similar architecture as $\mathrm{NN}_q^M$'s CNN (except that the filter size is 5) to process $x$. Then after the reshape-to-vector operation, the vector first gets transformed by an MLP of size $[4 \times 4 \times 64, 500]$, then it gets combined with $y$ and $m$ and passed through another MLP of size $[500 + \dim(Y) + \dim(M), 500, \dim(Z) \times 2]$ to obtain the mean and log-variance of $q(z|x, y, m)$. All the layers use ReLU activation except for the last layer, which uses linear activation.

- Measurement data experiments:

  – Discriminative DNN: The $A, C, X$ variables are concatenated to an input vector of total dimension 20. Then the DNN contains 3 densely connected hidden layer of $[64, 16, 32]$ width for each layer, and output $Y$. ReLU activations and dropout are used with dropout rate $[0.25, 0.25, 0.5]$ for each layer.

  – Deep CAMA's $p$ networks: we use $\dim(Y) = 5, \dim(A) = 5, \dim(C) = 5, \dim(Z) = 64$ and $\dim(M) = 32$.
    $p(y|a)$: an MLP of layer sizes $[\dim(A), 500, 500, \dim(Y)]$, ReLU activations except for the last layer (softmax).
    $p(x|y, c, z, m)$ contains 5 networks: 4 networks $\{\mathrm{NN}_Y^p, \mathrm{NN}_C^p, \mathrm{NN}_Z^p, \mathrm{NN}_M^p\}$ to process each of the parents of $X$, followed by a merging network.
    $\mathrm{NN}_Y^p$: an MLP of layer sizes $[\dim(Y), 500, 500]$ and ReLU activations.
    $\mathrm{NN}_C^p$: an MLP of layer sizes $[\dim(C), 500, 500]$ and ReLU activations.
    $\mathrm{NN}_Z^p$ an MLP of layer sizes $[\dim(Z), 500, 500]$ and ReLU activations.
    $\mathrm{NN}_M^p$: an MLP of layer sizes $[\dim(M), 500, 500, 500, 500]$ and ReLU activations.
    $\mathrm{NN}_{\mathrm{merge}}^p$: it first start from a concatenation of the feature outputs from the above 4 networks, then transforms the concatenated vector with an MLP of layer sizes $[500 \times 4, 500, \dim(X)]$ to output the mean of $x$. All the layers use ReLU activations except for the last layer, which uses linear activation.

  – Deep CAMA's $q$ networks:
    $q(m|x)$: it uses an MLP of layer sizes $[\dim(X), 500, 500, \dim(M) \times 2]$ to obtain the mean and log-variance. All the layers use ReLU activations except for the last layer, which uses linear activation.
    $q(z|x, y, m, a, c)$: it first concatenates $x, y, m, a, c$ into a vecto, then uses an MLP of layer sizes $[\dim(X) + \dim(Y) + \dim(M) + \dim(A) + \dim(C), 500, 500, \dim(Z) \times 2]$ to transform this vector into the mean and log-variance of $q(z|x, y, m, a, c)$. All the layers use ReLU activations except for the last layer, which uses linear activation.

- CIFAR-binary experiments:

  – Discriminative CNN: The discriminate model used in the paper is a CNN with 3 convolutional layers of filter width 3 and channel sizes $[128, 128, 128]$, followed by a flattening operation and a 2-hidden layer MLP of size $[4 \times 4 \times 128, 1000, 1000, 10]$. It uses ReLU activations and max pooling for the convolutional layers.

  – Deep CAMA's $p$ networks: we use $\dim(Y) = 10, \dim(Z) = 128$ and $\dim(M) = 64$.
    $\mathrm{NN}_Y^p$: an MLP of layer sizes $[\dim(Y), 1000, 1000]$ and ReLU activations.
    $\mathrm{NN}_Z^p$: an MLP of layer sizes $[\dim(Z), 1000, 1000]$ and ReLU activations.
    $\mathrm{NN}_M^p$: an MLP of layer sizes $[\dim(M), 1000, 1000, 1000]$ and ReLU activations.
    $\mathrm{NN}_{\mathrm{merge}}^p$: an projection layer which projects the feature outputs from the previous networks to a 3D tensor of shape $(4, 4, 64)$, followed by 4 deconvolutional layers with stride 2, SAME padding, filter size $(3, 3, 64, 64)$ except for the last layer $(3, 3, 64, 3)$. All the layers use ReLU activations except for the last layer, which uses sigmoid activation.

  – Deep CAMA's $q$ networks:
    $\mathrm{NN}_M^q$: it starts from a convolutional neural network (CNN) with 3 blocks of $\{\mathrm{conv}3 \times 3, \mathrm{max\text{-}pool}\}$ layers with output channel size 64, stride 1 and SAME padding, then performs a reshape-to-vector operation and transforms this vector with an MLP of layer sizes $[4 \times 4 \times 64, 1000, 1000, \dim(M) \times 2]$ to generate the mean and log-variance of $q(m|x)$. All the layers use ReLU activation except for the last layer, which uses linear activation.
    $\mathrm{NN}_Z^q$: first it re-uses $\mathrm{NN}_M^q$ CNN network for feature extraction on $x$. Then after the reshape-to-vector operation, the vector gets combined with $y$ and $m$ and passed through another MLP of size $[4 \times 4 \times 64 + \dim(Y) + \dim(M), 1000, 1000, \dim(Z) \times 2]$ to

obtain the mean and log-variance of $q(z|x, y, m)$. All the layers use ReLU activation except for the last layer, which uses linear activation.

**Measurement data generation**   We set the target $Y$ to be categorical, its children, co-parents and parents are continuous variables. The set 5 classes for $Y$, and $Y$ has 10 children variables and 5 co-parents variables, also one 5 dimensional parents.

Parents ($A$) and co-parents ($C$) are generated by sampling from a normal distribution. We generate $Y$ using structured equation $Y = f_y(A) + \sigma_Y$. We use $f_y = \text{argmax } g(A)$ and $g()$ is a quadratic function $0.2 * A^2 - 0.8A$. $\sigma_Y$ is the Gaussain noise.

To generate the children $X = f(Y, C) + \sigma_x$, we also used quadratic function $f$ and the parameters were sampled from a Gaussian distribution. As in the experiment, we were using fixed scale shift, we also added a normalize the children before adding the Gaussian random noise $\sigma_x$. So that all observations are in similar scale.

**Other**   For MNIST experiments, we uses $5\%$ of the training data as the validation set. We used the training results with the highest validation accuracy for testing. If not otherwise specified, $50\%$ of noisy test data are used for fine-tuning in the shift experiments and all data are used for fine-tuning in the attack experiments.

For the experiments with measurement data. We generated 1000 data in total. We split, 500 data for testing, 450 for training and 50 for validation. We used the training results with the highest validation accuracy for testing for both deep CAMA and for DNN.