[Reviews · NeurIPS 2020]

Review 1

Summary and Contributions: This paper discusses robustness to input manipulations in neural nets from a causal point of view. From this perspective, the authors motivate the design of a conditional latent variable model, whose latent space is explicitly decomposed into two parts, one is supposed to encode underlying (nuisance) factors that can be intervened, and the other one is intended to represent the remaining factors, which cannot be manipulated. A fine-tuning method on test data is further proposed to improve robustness to unseen manipulations. Empirical results show that the proposed model achieves the desired disentanglement in the latent space, and it is more robust compared to a deep neural net baseline.

Strengths: Building robust classifiers to input manipulations is an important topic. The proposed method is technically sound, and its benefits as compared to a discriminative deep neural net baseline are supported by numerical experiments.

Weaknesses: The technical contribution of this paper is marginal. The proposed model is a form of conditional variational autoencoder (CVAE) [1, 2]. Perhaps one difference with some existing CVAE variants (e.g., p(z,x,y) = p(z)p(y)p(x|z,y)) is the explicit decomposition of the latent representation into two parts Z, M, with the latter encoding nuisance factors, which can be intervened, and the former representing the remaining factors, which cannot be intervened. However, it is not clear (without comparisons with CVAE) if such decomposition is necessary to achieve the desired robustness, since VAE models can be good at learning disentangled representations. Moreover, the causal treatment and the introduction of the “do” notation do not seem to be necessary, making the positioning of this work from the causal perspective a bit weak. Experiments are weak in that the proposed model is compared to a relatively weak baseline, namely a discriminative deep neural net. Stronger baselines should be considered (e.g., Conditional Variational Autoencoder, Deep Variational Information Bottleneck (IB) [3]) to assess the importance of the proposed model. Some relevant references are missing [3, 4]. [1] Sohn, Kihyuk, Honglak Lee, and Xinchen Yan. "Learning structured output representation using deep conditional generative models." NeurIPS. 2015. [2] Siddharth, Narayanaswamy, et al. "Learning disentangled representations with semi-supervised deep generative models." NeurIPS 2017. [3] Alemi, Alexander A., et al. "Deep variational information bottleneck." ICLR (2017). [4] Moyer, Daniel, et al. "Invariant representations without adversarial training." NeurIPS. 2018.

Correctness: Yes to the extent that I checked.

Clarity: The writing of the paper is fine.

Relation to Prior Work: Relation to conditional variational autoencoders can be improved. Please refer to the weaknesses section for more details.

Reproducibility: Yes

Additional Feedback: Under the current objective, the latent variable Z may still encode some nuisance information, which can be intervened. It would be interesting to consider encouraging Z and M to be independent explicitly. Typo: Is significantly more robustness *** Post Rebuttal Update *** I acknowledge having read the authors rebuttal as well as the other reviews, and I decided to keep my original assessment unchanged for the following reasons. The positioning of the paper from the causality perspective is weak, the introduction of the “do” notation seems unnecessary and can be replaced by the usual observational conditioning. The model derived in 3.1 is a conditional VAE (CVAE) with a decomposed latent space. While the authors also proposes an interesting fine-tuning method, and extend their model to the multimodal case, the model of section 3.1 is central in this work. I would therefore recommend providing theoretical/empirical justification on the importance of the decomposition of the CVAE’s latent space, which I suspect may not be necessary to achieve the desired robustness. The only considered baseline is relatively weak in that it is deterministic while the proposed model is probabilistic. The rebuttal includes some results of CVAE and DVIB (Deep Variational Information Bottleck). While on the PGD attacks the proposed method seems to outperform the above two baselines, on “the vertical shift range” the results of CAVE, DVIB and the proposed CAMA are very tight, justifying the importance of considering stronger baselines. Moreover, I would recommend applying the proposed fine-tuning procedure to CVAE and DVIB for more fair comparisons.


Review 2

Summary and Contributions: This paper proposes a causal approach to building latent representations of data robust to manipulations unrelated to the downstream task. The proposed method takes advantage of the knowledge of manipulations present in training to various degrees. Furthermore, their method supports fine-tuning at test time to adapt to unseen manipulations during training.

Strengths: The goal of this paper is ambitious and steps beyond the assumptions made with traditional statistical learning methods. In particular, the ability to make correct predictions on out of distribution data. The causal graph proposed in this work assumes independence between the downstream task and the manipulation, which, despite being very simplistic, also seems to be very reasonable. In general, manipulations, are not going to impact the prediction label by definition. Through experiments, the authors demonstrate improved robustness in out of distribution shifts for MNIST and adversarial attacks. Importantly, this is without explicit knowledge of the attacks or domain shifts. The manipulation prior proposed in this work seems very general and could apply broadly to more datasets than methods that explicitly address known dataset shifts.

Weaknesses: Since the authors demonstrate robustness in a non-standard domain (which is a good thing), the authors should do a better job of presenting what exactly must be known about the given dataset during training compared to a standard dataset. While exact knowledge of the domain shift is unknown beforehand, there is the knowledge that manipulation has occurred. For example, what is the exact format of the training data in line 205? Are corresponding pairs of manipulated and manipulated inputs given? Or is it just the binary label distinguishing whether the data came from the manipulated set or clean set? Equation 5 would imply the latter, but this needs to be clarified because the introduction (lines 36-37, 41-43) describes shortcoming of standard DNNs for unseen causes. However, this model does not appear to address this issue without having access to additional information. Such a clarification would help readers understand what types of data this method is suitable for and highlight how this method leverages this additional information.

Correctness: The claims appear correct, though this reviewer is not familiar with causal graphical models.

Clarity: Aside from the ambiguity of what an unseen manipulation means in the context of the introduction compared to the training datasets used in the experiments, the paper is well-written.

Relation to Prior Work: This work does mention how the method considers unseen manipulations, whereas past work does not. As mentioned in an earlier section of this review, this needs to be clarified better to distinguish it more precisely from work like: Arjovsky, Martin, et al. "Invariant risk minimization." arXiv preprint arXiv:1907.02893 (2019). which is mentioned in their related work as not considered 'unseen' manipulations. The level of knowledge of the IRM approach appears comparable to the method proposed here. IRM only requires knowledge that manipulation has occurred, not information about the specific manipulation.

Reproducibility: Yes

Additional Feedback: ----After Rebuttal---- After going through the rebuttal and other reviews, I will keep my current score of this review. My main concern about how much knowledge of the manipulation is presented to the model during training is addressed the rebuttal. But I would suggest the authors make this knowledge very clear in future revisions. This will be a point of comparison and contrast with other work. Furthermore, fine-tuning at test time paradigm to handle out of distribution data is an interesting paradigm that I hope to see become more common.


Review 3

Summary and Contributions: The paper proposes an inference framework, deep CAMA, that takes into account the causal relationship between the data and effect variables. In the framework, an encoder and generative model learn the relationships that link data samples (X) to labels (Y) and effect variables (M and Z) where M can be changed by a malicious party during adversarial attacks. During inference, the model parameters can be finetuned to learn unseen M intervention to produce robust Y predictions. The approach is interesting, especially the finetuning phase which is novel to the best of my knowledge. Though the experiments are somewhat limited to small-scaled tasks here and the adversarial robustness performance is still far from the best defenses, this work may inspire more future defenses which infer inputs with a causal view. One main concern I have is regarding the scalability of the approach versus other defenses such as adversarially trained discriminative models. Also, for the cases of horizontal/vertical shift, I am guessing the discriminative model could be robust to both types of shift by training on both horizontal/vertical shifted data. It would be more compelling if deep CAMA can outperform the discriminative model trained with this data augmentation scheme. ========================================= I acknowledge that I read the rebuttal and thank the authors for addressing the questions and concerns I had.

Strengths: - The paper shows that Deep CAMA can confer robustness to some unseen perturbations such as translational shift and may inspire future defenses taking the causal approach to robustness. - The proposed finetuning phase to learn unseen M intervention is novel and sets this paper apart from previous work like Narayanaswamy et al.

Weaknesses: - Experiments are mostly conducted on toy tasks - Unsupported and imprecise statements in the paper. (see more on clarity below)

Correctness: Yes

Clarity: The paper is generally easy to follow but has unsupported/exaggerated statements such as: - Line 52: “They simply trust the observed data and ignore the constraints of the data generating process, which leads to overfiting to nuisance factors that do not cause the ground truth labels." Also, important details such as how Deep CAMA predicts its outputs given input data are not clearly stated in the paper.

Relation to Prior Work: Yes, the proposed finetuning phase to learn unseen M intervention is novel and sets this paper apart from previous work like Narayanaswamy et al.

Reproducibility: Yes

Additional Feedback: - What is the total parameter size of Deep CAMA versus the discriminative models in your experiments? - Improve clarity in how Deep CAMA infer input data, e.g. how does the Monte Carlo approximation to Bayes’ rule work? minor edits: Line 49: “is significantly more robustness to unseen manipulations.”, robustness>robust


Review 4

Summary and Contributions: This paper (A Causal View on Robustness of Neural Networks) defines a model for causal modeling of manipulations, for improving robustness of decision making methods. The importance of introducing causality framing into this problem is shown, a new mdeol (deep CAMA) is introduced, and a number of experiments are used to show the benefits CAMA can provide.

Strengths: The proposed model, while fairly involved, manages to both explain the core design principles, and necessary background to understand the design, as well as the functionality of the introduced CAMA method. Doing so in a limited space is not easy, and I commend the authors on the description of their approach as well as the necessary background. The experimental section, while performed on limited datasets, is thorough in the types of experiments performed and the breadth of use of the data itself. The inclusion of reproduction code for the model and experiments in the supplementary material is good, and I strongly encourage the authors to release it publicly upon (potential) publication. Doing so will allow many researchers to deeper understand and extend on the work shown here.

Weaknesses: Demonstrating the method on more involved image datasets than MNIST / CIFAR-10, other image tasks, or in more domains (audio separation, dereverberation, and other audio scene tasks are a place where causal knowledge could potentially boost performance) would be beneficial. However, the limited datasets are countered by multiple detailed experiments upon these datasets, so this is not a huge detraction from the paper, especially given the core contribution of the model itself. Describing roughly the practical cost of CAMA (in terms of computation) would give an idea of the robustness/computational performance tradeoff for some of these tasks, which would be useful information for practical usage.

Correctness: The derivation and core concepts of CAMA appear to be correct. The experimental methodology taken to test various aspects of deep CAMA is well-motivated, and the experiments show the benefits of using the model compared to baselines without using CAMA.

Clarity: The paper is well written, especially given that it introduces a fairly complex model, using a number of tools from causal modeling, and manages to succinctly define the model itself, the tools necessary to understand the model design, and a number of experiments within the page limit.

Relation to Prior Work: A number of related works on adversarial robustness are described, and how CAMA "fits in" to the larger picture of models attempting to handle systematic generalization and adversarial robustness is clear given background and reference material.

Reproducibility: Yes

Additional Feedback: Given fundamental limits of network robustness to adversarial attacks (see "Limitations of Adversarial Robustness: Strong No Free Lunch Theorem"), where does the proposed method differ, or relate to that general framework for robustness / adversaries? Does the causality framework provide a "way out" from the bounds and limits shown in that work? The lack of robustness to horizontal and vertical shift in the MNIST example seem as coupled to the architectural bias of the particular discriminator design, as to the task itself - for example an object detection framework such as RCNN or modern variants (ala Mask-RCNN) should have little issue with the shifted image task described in the paper. How can we separate the issue of network design (which is frequently driven by known invariances in the desired domain - such as moving from simple DNNs to more applicable CNNs) and the causal manipulation model (which also has design parameters and potential pitfalls, as discussed in 3.2 and 4.2). If using some kind of automated network design setting (such as meta-learning or evolutionary approaches) would both the CAMA model design, and the discriminator itself need to be designed in conjunction, or some kind of back-and-forth iteration? Does knowing something about the CAMA model potentially tell you something about the ideal design of the classifier as well? Just before section 5, the statement "deep CAMA remains to be more robust than the discriminative DNN when the mis-specification is not too severe". Are there measures to determine mis-specification post-hoc? Are there guidelines or principles for model design to limit or avoid mis-specification in practice? After feedback: Thank you to the authors for descriptions and clarifications in the feedback document, and for clarifying the questions I raised during the review. The comments and discussion with regards to all reviewers should further strengthen this paper, and the discussion was very helpful.

[Author Response · NeurIPS 2020]

1 We thank the reviewers for their time and insightful comments on our paper. We respond to each reviewer(R) below. In
2 addition, we will make the code publicly available, together with the paper.

3 R1. The proposed model is a form of conditional variational autoencoder (CVAE) : We believe this is a misunderstanding,
4 and our contribution goes far beyond that. Our contributions are in three-fold: (1) a causal view on the robustness of
5 neural networks and the discussion on valid artificial manipulations; (2) the CAMA model as an instance of causal
6 consistent generative models, including versions for both single modality data and generic causal graphs; (3) fine-tuning
7 approaches to improving model robustness to unseen manipulations. R1's comments cover only part of contribution (2),
8 i.e., the CAMA model for single modality data. R3 also pointed out "The proposed fine-tuning phase to learn unseen M
9 intervention is novel and sets this paper apart from previous work like Narayanaswamy et al." We discussed in detail
10 how our work differs from CVAE-type of models in appendix B, and will clarify on this further in revision.

11 R1.decomposition necessity and CVAE comparisons: We emphasise that CVAE and other mentioned work can
12 only be applied to single modality data, which is a special case discussed in our paper. For empirical comparisons,

we present on the left of below figure the robustness of CVAE
(Sohn et al. [1]) & DVIB (Alemi et al. [3]) to vertical shifts
and against PGD attacks, which clearly shows the advantage of
CAMA, especially the fine-tuning ability. Similar results have
been reported in [25] of our paper's reference, and our work
provides extra advantages due to the use of a causally consistent
model and the fine-tuning method motivated by causal reasoning.

20 R2. training data format... additional information: The training procedure only requires knowledge on whether the
21 training data is clean (in such case we set $m = 0$) or manipulated (potentially with unknown manipulation). In the
22 manipulated case we do not require explicit label for $M$ and perform inference on it instead. For test data, the labels for
23 both $Y$ and $M$ are not available, only the input X is given, similar to the standard test setting.

24 R2. Comparisons to IRM: First, IRM only considers single modality data. Moreover, IRM aims to find representations
25 that are invariant to environmental changes, which does not necessarily provide robustness to *unseen* manipulations.
26 Imagine training a model with IRM using clean MNIST and MNIST shifted $50\%$ to the left, then the model is very
27 likely to fail when tested on MNIST shifted $50\%$ to the right. Will clarify in revision.

28 R3. (adversarial) data augmentation: Deep CAMA also benefits from adversarial training (Figure 10). Both CAMA
29 and discriminative model are robust to *known* adversarial attacks that are observed in adversarial training. However, the
30 key advantage of CAMA over discriminative models is in its robustness to *unseen* manipulation. First, Figure 1 shows
31 that training discriminative DNNs with one manipulation can hurt its robustness to another unseen manipulation. This
32 is not the case for CAMA (see the overlapping green & orange curves in Figure 7(b)). Second, with fine-tuning (which
33 is another main contribution of our work), CAMA can be significantly more robust to unseen manipulations.

34 R3. concerns on line 52 statement: In causality literature, causal factorizations correspond to modular/independent
35 mechanisms [34,38] of our paper's reference; building ML models consistent with these causal factorizations also
36 improves robustness [19] of our paper's reference. Will clarify further by expanding the arguments.

37 R3. Bayes' rule prediction clarity: As shown on the RHS of eq. (6), we sample $m$ from $q(m|x)$ and sample $z$ from
38 $q(z|x,y,m)$ with the previously sampled $m$. In the paper, we use 1 sample for $m$ and $K$ samples of $z$ associated with
39 each $m$ sample. Given the sampled $m$ and $z$ instances, we can compute the terms inside the log for each $y = c$ (as an
40 approximation to $p(x, y = c)$), and apply softmax to obtain the predictive distribution. Will clarify further on it.

41 R3. parameter size: The network size detail is presented in appendix D for all experimental settings. In appendix C
42 (Figure 16), we present the performance of discriminative models with larger network sizes and the result is similar to
43 the one with smaller network sizes.

44 R4. computation cost: The time complexity of CAMA is in the same order of a regular VAE. For predictions, fine-tuning
45 requires a small amount of additional time, as only a small fraction of data is needed for fine-tuning (Figures 8 & 20).

46 R4. "Limitations of Adversarial Robustness": We do not intend to claim CAMA's robustness to all possible manipu-
47 lations *straightaway after training*. So the theory of the mentioned work (and many others) is applicable to CAMA
48 before fine-tuning. However, this line of theoretical work evaluates the robustness of a classifier trained on clean data
49 only. Rather, CAMA brings extra advantages by fine-tuning, which enables adaptation to the unknown manipulation in
50 test time.This adaptation is required whenever a new manipulation is present, allowing CAMA to "learn" to be robust to
51 a growing number of manipulations.

52 R4. Network architecture bias: Indeed RCNN-like networks can be more robust to shifts by design. However, Fig. 15
53 & 16 shows that the use of CNN (instead of MLP) does not fully address the over-fitting issue to seen manipulations.
54 Moreover, architecture design typically provides robustness to specific manipulations. Instead, our goal is to make the
55 model robust to (infinity number of) unseen manipulations, which cannot be archived by architecture design only.

56 R4. limit mis-specification: In practice, the causal graphs are either provided by domain experts or obtained by running
57 causal discovery algorithms. For the former, it requires working closely with domain experts to refine the causal
58 hypotheses. For the latter, choosing a suitable causal discovery algorithm for the application at hand would be critical,
59 and there exist approaches to evaluating the performance of causal discovery algorithms.

[Meta-Review · NeurIPS 2020]

The reviews were mixed. On one hand, the manuscript is well-organized and reviewers appreciated the probabilistic attempt at robustness and the fine-tuning idea. On the other hand, concerns were voiced in the reviews and during discussion. In the end, the meta-reviewer (after independent examination of the manuscript) concluded that the merits outweigh the potential issues. We strongly encourage the authors to revise the draft by taking the following comments into account (more in the reviews): (a) The causal aspect of the manuscript appears to be somewhat decorative than necessary. Indeed, upon independent reading of the manuscript, the meta-reviewer agrees that one can essentially remove all causal notions, after all the do-calculus on the simple model that the authors adopted is nothing different from the usual conditioning. Besides, the causal model is never used for true intervention or performing counterfactual inference. As Reviewer 1 pointed out, the proposed approach is essentially a VAE with factorized latent variables, and the authors should have compared to many existing VAE alternatives. Whenever possible, please choose the simplest formulation of your work and refrain from borrowing terminology to just make things look fancier without substance. If there is a true need of causal reasoning, then this needs to be better articulated and demonstrated. (b) The experiments were well performed in the sense of verifying the authors' approach but fell short in comparing to existing attempts at improving robustness, just to name a few (more can be easily googled): - Parseval networks - L_2-nonexpansive neural networks - Efficient Defenses Against Adversarial Attacks - Deep defense: Training dnns with improved adversarial robustness Comparison against at least some of these existing approaches should be included. A related concern is that the reason that the proposed algorithm appears to be more robust (than a straightforward dnn) may simply be because its clean accuracy is too low. As mentioned in the appendix, the proposed algorithm achieved less than 50% on CIFAR10, which is probably even worse than classic algorithms such as logistic regression or SVM. It is plausible that as accuracy increases, a model (such as a deep net) becomes increasingly fragile against adversarial examples. The authors need to rule this possibility out (the experiments on MNIST is not sufficiently convincing as MNIST is known to be quite easy). Put it another way, if a practitioner adopted the authors' model, then an adversary need not do anything at all but is already able to drive the accuracy to lower than 50% on CIFAR10. (c) During prediction (eq (6)), the algorithm needs to sample m and z for each possible label. This is essentially a Bayesian approach to impute the unobserved variables. How sensitive is the final result wrt this Monte Carlo sampling? It is well-known that the Bayesian approach offers some robustness by averaging over a number of models (see McKay or Bishop's book). Could the authors explain their approach through this well-known principle? There is nothing free though: One would need more training data to train the latent variable model. This might explain why the authors were not able to achieve decent accuracy on CIFAR10. The trade-off between clean accuracy and robustness in a latent variable model needs to be better illustrated.